# Development of an Abalone 3D Food Printing Ink for the Personalized Senior-Friendly Foods

**DOI:** 10.3390/foods11203262

**Published:** 2022-10-19

**Authors:** Hyun-Jung Yun, Na-Ra Han, Hyun-Woo An, Won-Kyo Jung, Hyun-Woo Kim, Sang-Gil Lee

**Affiliations:** 1Research Group of Food Processing, Korea Food Research Institute, Wanju 55365, Korea; 2Department of Food Science and Nutrition, College of Fisheries Science, Pukyong National University, Busan 48513, Korea; 3Department of Smart Green Technology Engineering, Pukyong National University, Busan 48513, Korea; 4Research Center for Marine Integrated Bionics Technology, Pukyong National University, Busan 48513, Korea; 5Department of Biomedical Engineering, Pukyong National University, Busan 48513, Korea; 6Department of Biotechnology, College of Life Science and Biotechnology, Korea University, Seoul 02841, Korea

**Keywords:** 3D food printing, foods for elderly, gelatin, printability

## Abstract

Notably for seniors, 3D food printing is an appropriate processing method for creating customized meals that meet their unique nutritional requirements and textural preferences. This study attempted to develop an ink for food 3D printers containing abalone powder and several nutrition properties that meet the criteria for senior-friendly foods. The texture of the products was adjusted using gelatin. The ink consisted of abalone powder (10%), soybean protein (4.5%), polydextrose (2.5%), vitamin C (0.0098%), and gellan gum (1%). To examine the physicochemical properties of the ink, texture, water holding capacity, and rheological properties were measured. In addition, the suitability of the 3D printing was examined. As a result, 3% gelatin 3D food printing ink demonstrated optimal printability and could be converted into foods that could be consumed in one step (teeth intake), depending on the types of food for seniors.

## 1. Introduction

In Korea, the senior population is growing quickly as a result of declining birth rates and rising life expectancy; in 2020, it was 15.8%, and by 2060, it is expected to rise to 43.9% [1]. Physical changes due to aging include loss of natural teeth, weakening of coordination of body movements, and weakening of mastication muscles [2]. As a result, symptoms, such as poor appetite, decreased chewing, and swallowing function lead to nutritional imbalance. Although it is urgent to develop foods for the elderly due to an aging society, Korea’s elderly foods market is not active due to problems such as lack of awareness, business feasibility, marketability, planning, and technological development compared to global markets [3]. Following global trends, in 2019, the Korean Ministry of Agriculture, Food and Rural Affairs announced Korean industrial standards (KS certification) of senior friendly food to encourage the manufacture of senior-friendly food; fulfilling three standards among nine nutritional standards and one physiological property (amounts of protein, dietary fiber, vitamin A, C, D, riboflavin, niacin, calcium, and potassium).

Supplying an appropriate concentration and texture of foods is important to the elderly, but food that has the texture modified may lose its appearance and taste. 3D printing is a suitable processing method for personalized food production because it can adjust physical properties by applying various patterns to foods, and nutrients can be supplemented by controlling the amount and nutrition of food. In addition, 3D printing enables customized meal production based on individual specific nutritional needs and calorie intake and is suitable for people with specific nutritional needs, such as the elderly or patients who have difficulty eating or swallowing [4,5]. Therefore, personalized textures can be produced through material density control and changes in printed internal structure using 3D printing can lead to the development of elderly foods [6].

Applying 3D printing in the food industry is important to select a suitable printing technology because of the various physicochemical properties of food ingredients. There are several techniques used in 3D food printing, including extrusion-based printing, selective laser sintering, and binder jetting. In the extrusion-based 3D printing which is used in this study, in order to ensure smooth printing, the fluidity of the food should be maintained in the pre-extrusion stage and the shape should be maintained without structural deformation when extruding material. The appropriate strength of samples in the post-extrusion stage is important [4,7]. Therefore, 3D printing ink should have uniform particles and proper flow characteristics during printing, and be able to provide structural stability after printing [8].

Gelatin is a type of hydrogel made by hydrolyzing protein in collagen and has remarkable physical properties such as dispersion stability and moisture retention [9]. Gelatin is frequently used as a gelling agent and a stabilizer to provide texture properties for food components [10]. 

In addition, gelatin is widely used as a 3D printing material due to its good hydrogel formation properties [11], and can be deposited after extrusion from a syringe to form a shape [12]. However, gelatin gel is formed by intermolecular contact of hydrogen bonds, but the triple helical structure returns to a twisted state at 37 °C and melts in solution form [13,14], which is greatly affected by temperature and concentration during 3D printing. Mixing gelatin and a small amount of gellan gum improves gelatin properties such as gelation, melting temperature, gel strength, and minimizes thermal denaturation properties.

Abalone is a high-protein food containing 15 g of protein per 100 g and high nutrients and physiologically active compounds to promote health [15]. The protein of abalone is one of the most important nutrients in the foods of the elderly, as large amounts of protein intake are effective in preventing various diseases and muscle mass losses caused by aging, improving bone health, maintaining energy balance, cardiovascular function, and wound healing [16]. In addition, abalone is an abundant marine resource in Korea and can be used in various food industries [17]. However, the hard and chewy texture of abalone makes it difficult for elderly people to intake and digest, and research on 3D printing ink related to elderly foods containing abalone is insufficient.

Therefore, this study proposes 3D food print ink containing abalone which can be adjusted in texture according to gelatin concentration for the elderly who have difficulty in consuming seafood and masticatory function. The results of this study are intended to show the printability and personalized elderly food processing of 3D printing according to gelatin content. Texture analysis and other rheological experiments are conducted to confirm the criteria for elderly food properties.

## 2. Materials and Methods

### 2.1. Materials

*Haliotis discus hannai* (abalone) was purchased from a local market (Busan, Korea). Isolated soy protein was obtained from Solae Co. (St. Louis, MO, USA). Gellan gum (CP Kelco U.S., Inc., Atlanta, GA, USA) and ascorbic acid (Sigma-Aldrich Co., St. Louis, MO, USA) were used. Polydextrose and gelatin (from porcine skin, type A) were purchased from Samyang Corp. (Seongnam, Korea).

### 2.2. Abalone Powder

Abalone was washed with tap water and vacuum-sealed using a vacuum sealer (Solis vac smart type 577, Glattbrugg, Switzerland). The packed meat was cooked in an 80 °C water bath for 30 min with a sous-vide machine (ANOVA precision cooker, Anova Applied Electronics, Inc., San Francisco, CA, USA). The sample was sliced and freeze-dried for 72 h at −70 °C. The dried abalone was milled using a blender, and abalone particles were sieved on a sieve shaker (LAWSON Scientific, Hangzhou, China) with stainless steel sieves (Chunggye sieve Co., Ltd., Seoul, Korea) with pore sizes of 200 μm. After sieving, the abalone powder (AP) was stored at −50 °C until further use.

### 2.3. Preparation of the 3D Print Ink with Abalone (Abalone 3D Print Ink: API)

The powder mixing ratio is based on the nutritional standards of elderly foods. About 10 g of abalone powder was mixed with isolated soy protein (4.5 g) for protein enhancement, polydextrose (2.5 g) for dietary fiber fortification, vitamin C (0.0098 g), gellan gum (1 g), and gelatin (0, 1, 3, 5, and 7 g) to prepare the mixture (Table 1). Especially, although isolated soy protein sometimes causes allergic reactions, it was used for protein enhancement as one of the commonly used protein food additives. The samples were named G0, G1, G3, G5, and G7 according to gelatin concentrations (0, 1, 3, 5, and 7 g). Distilled water (dw) of 55 °C and swelled gelatin at ice dw were added to each mixed powder. Abalone 3D print ink (API) was stored at 4 °C and used within 48 h.

### 2.4. Texture Analysis

A texture analyzer (FRTS 50N, IMADA Co., Ltd., Jinnoshinden-cho Knowari Toyohashi, Japan) was used to measure the texture properties of API. For the compression test, a 20 mm diameter of the circular probe was used, and the compression and return speed were 10 mm/s. The APIs were filled in cylindrical containers (diameter of 40 mm, height of 20 mm) and compressed to 5 mm of height. These test conditions were based on the standards of the Ministry of Food and Drug Safety. All tests were conducted at least three times.

### 2.5. Water Holding Capacity (WHC)

The water holding capacity of APIs with different concentrations of gelatin was determined using the method described by Le et al. [18]. In addition, 500 mg of each API was transferred to a tube, and the weights of the samples were measured before centrifugation (W1). The tubes containing the sample were centrifuged (1730R, Labogen, Korea) for 20 min at 14,000 rpm. The supernatants were removed, and the tubes containing the residue (W2) were weighed. The following formula was used to calculate the WHC:WHC (%) = (W1 − W2/W1) × 100

### 2.6. Rheological Properties of API

Before printing, the rheometer (MCR 92, Anton Paar Inc., Graz, Austria) was used to confirm the rheological properties of API. A steel flat plate (diameter of 25 mm, gap of 1 mm) was used in all tests, and samples were placed on the plate. The strain sweep test was conducted at room temperature, and the strain range was 0.1–100% kept at a fixed frequency of 10 rad/s for determining the linear viscoelastic (LVE) area of API. The shear strain value was determined to be 0.3%, and all subsequent experiments were performed within the LVE of the samples. 

The frequency sweep tests were performed in the range of 0.1–100 rad/s at room temperature. Storage modulus (G′), loss modulus (G″), and complex viscosity (η*) were calculated using RheoCompass^TM^ software (Anton Paar).

### 2.7. 3D Printing Process

The 3D food printer (YOLILAB 1.0, Yolilo Co., Seoul, Korea) and 60 mL of the syringe with nozzle tip (diameter of 1.1 mm) were used for printing. The 3D printed model was a cylindrical structure (diameter of 28 mm, height of 40 mm) and was converted to G-code using 3D printing slicing software Cura 4.10.0 (Ultimaker B.V., Geldermalsen, The Netherlands). The conditions of the 3D printing process were as follows: room temperature (25–27 °C), height of the first layer of 1.2 mm, height between the layer and the layer of 0.9 mm, moving speed of 25 mm/s, infill density of 0%, vertical shells of 20 lines, and z offset of 0.5 mm. 

The samples were placed at room temperature for an hour to equalize the pastes of API and filled into the syringe. The cylindrical shape was used to confirm the printability of API and elaboration of the model. To compare with the set 3D model and printing structure, the diameter and height of the printing were measured, and the printing performance of API produced according to gelatin concentration was confirmed.

### 2.8. Scanning Electron Microscopy (SEM)

Microstructures of the sample were observed to confirm structural characteristics under gelatin concentrations using scanning electron microscopy (SEM) (S-2400, Hitachi, Tokyo, Japan). The freeze-dried APIs of cross-sections were fixed on carbon tape. The fixed samples were coated with gold, and images were captured at 5 kV with 200× magnification.

### 2.9. Statistical Analysis

All experiments were carried out in triplicate, expressed as mean ± standard deviation, and analyzed by one-way analysis of variance (ANOVA) with Tukey’s test in Prism 9.0 (Graphpad Software Inc., La Jolla, CA, USA) (*p* < 0.05).

## 3. Results

### 3.1. Texture Analysis

The effects of the gelatin concentrations on textural properties are shown in Figure 1. Hardness implies the degree of food required to compress the food into an ingestible form between the teeth or the tongue [19], and it is a major factor that divides the food stage into three in terms of elderly food. Hardness measurements were performed to confirm the food level to determine the suitability of APIs to elderly foods based on gelatin content.

The APIs according to gelatin concentration showed a significant increase in hardness, and the elderly food standards also showed differences. The results of G0 (6.6 ± 2.1 kN/m^2^) and G1 (15.4 ± 1.3 kN/m^2^) were under 20 kN/m^2^ and came out as the third level of elderly food standard; G3 (54.5 ± 4.3 kN/m^2^), G5 (85.2 ± 2.9 kN/m^2^), and G7 (107.4 ± 4.1 kN/m^2^) were in the more hardness stage (first level, which is over 50 kN/m^2^). All APIs containing 1 to 7% gelatin met the criteria for the properties of elderly foods. 

The gelatin solution becomes more elastic during the gelation process because irregular ring-shaped gelatin cooled and formed a three-dimensional structure by cross-linking of the polypeptide chains that try to return to the original triple helical structure [20]. As the gelatin concentration increases, the density of the crosslinking increases and forms a rubber network with strong elasticity [21,22]. According to Almeida and Lannes [23], gel strength varies between 3.33 to 6.67% depending on how much gelatin was extracted from the chicken; the higher the concentration, the stronger the gel. Supavititpatana et al. [24] confirmed that corn-added yogurt improved in hardness as the gelatin concentration increased (0, 0.2, 0.4, and 0.6%). This finding demonstrated the suitability of gelatin for regulating the texture of senior-friendly cuisine.

### 3.2. Water Holding Capacity

Gel made of food can cause moisture loss due to storage and temperature, which may affect the texture and quality of the gel. Therefore, the evaluation of food gel suitability is important through WHC, the ability of the sample to retain water when there is no external force [25,26].

The WHC of the API according to gelatin concentration was shown in Figure 2. The WHC of the API increased significantly with the increase in gelation addition, indicating that the gel network holding water was improved. There was no significant difference between G3, G5, and G7, but significantly higher WHC was confirmed compared to G0 and G1.

Due to its ability to absorb moisture, the enlarged gelatin network with the increase in gelatin concentration displayed high WHC [27]. The WHC of API was enhanced by the addition of gelatin, but the WHC of G0 was reduced because there was no gelatin present and no cross-linking of the protein as a result.

### 3.3. Rheological Properties of API

The food ink of the extruded 3D printing was deposited through the nozzle by the force exerted by the hydraulic piston, and the rheological properties of the material used for the ink affect the 3D printing [8]. Figure 3A shows the linear viscoelastic region (LVE) according to the increase in strain. Abalone 3D print ink deformation occurred reversibly within the LVE area but exceeded LVE causes destruction of the sample structure [28]. A storage modulus (G′) signifies the structural strength and elasticity of the gel, and loss modulus (G″) denotes viscosity characteristics [29]. According to the results of the amplitude sweep test, all samples appeared as LVE at 0.3% strain, and this strain was applied to all subsequent experiments.

G′ of all samples was higher than G″, and there was no intersection point (Figure 3C,D), which indicated that a material forming structure such as a gel was in an elastic state [30]. It was observed that the API material maintained its shape after 3D printing. In addition, as frequency increased, G′ and G″ increased in all samples (Figure 3C,D), suggesting that it was affected by gelatin concentration. Choi and Lim [22] reported that the higher the gelatin concentration, the higher the storage modulus, which was similar to the results presented in this study.

G5 and G7 show similar G″ values according to the increase in frequency (Figure 3D), and it can be predicted that the viscosity characteristics of the two samples were almost the same. G0 and G1 have similar G′ values without variation at low frequency (0.1–1 rad/s) (Figure 3C), suggesting that the sample remained unchanged in the stationary state and the gel-like properties of the two samples were similar.

All samples showed a shear-thinning phenomenon in which complex viscosity decreased as frequency increased (Figure 3B), meaning that all API samples have rheological features suitable for 3D printing. Shear-thinning behavior indicated that the polymer material or concentrated dispersion of food containing a long chain decomposed to align the food structure. Foods with shear-thinning behavior can be printed smoothly to the nozzle due to the high shear force of the extruder in the syringe, and the shape of the 3D-printed food was maintained after printing when the shear force was not applied [7]. 

### 3.4. 3D Printing Process

Abalone 3D print inks according to gelatin concentrations were 3D printed in a cylindrical shape (diameter of 28 mm, height of 40 mm) (Figure 4A). The 3D printing test was conducted to confirm the support of the structure and continuous printing of food ink.

G0 containing no gelatin sometimes was not outputted during printing, and the cylinder was also tilted. Compared to the G0, the G1 showed better printability but there were some parts where the sample was not printed, and the sophisticated printing was still difficult. However, the G3 was able to continue printing, and all subsequent printings were printed with accuracy and replicating the original model. This result showed a similar tendency to Figure 1 and Figure 3; therefore, it could be said that the improvement in mechanical strength was due to the increase in gelatin concentration.

Figure 4B shows the result of measuring the diameter and height of each cylinder. G3 (diameter of 28.33 mm, height of 39 mm) was confirmed the closest to the Con cylinder and indicated no significant difference with Con. Cylindrical height showed no tendency according to gelatin concentration, and G3 has no significant difference from Con. In addition, the diameters of G1, G3, G5, and G7 were not significantly different from the Con.

### 3.5. Scanning Electron Microscopy (SEM)

The microstructure of food is closely related to the texture, which varies depending on the properties of the ingredients [31]. Scanning electron microscope images of API according to gelatin concentration were presented in Figure 5. G0 and G1 showed a coarse surface and fragmented texture. Samples containing less gelatin had a non-uniform cross-section due to the lack of gelatin to bind powders or connect the materials, which was similar to the 3D printing results (Figure 4).

On the other hand, G3, G5, and G7, the higher concentrations of gelatin, have circular holes and thick walls. The thicker walls were observed with increasing gelatin concentrations (3–7%); this was a consistent microstructure with pure gelatin according to concentration [32]. In addition, walls of tissue thickness have a similar tendency to the hardness of texture (Figure 1). Skopinska-Winsniewska et al. [33] confirmed that, as the gelatin hydrogel matrix increased, the thickness of the microstructure walls increased and the texture became harder.

## 4. Conclusions

This study reported the invention of 3D print ink for elderly foods based on gelatin content, which influences texture. The third level of geriatric food standard was API with 0% and 1% gelatin, but due to its unstable structure and winding lines, it was inappropriate for 3D printing. Three percent or more gelatin was added to API to make it appropriate for 3D printing ink at the first level of the standard. Increasing gelatin concentration can improve the 3D printing performance of food ink. The textural properties of food ink could be adjusted according to gelatin concentration. In addition, the rheological characteristics of the ink, the 3D printing process, and microstructure results showed the same tendency. Therefore, the information presented in this study confirmed that the 3D food ink for the elderly is capable of controlling texture and may be applied to the development of elderly foods and the application of 3D printing technology.

## Figures and Tables

**Figure 1 foods-11-03262-f001:**
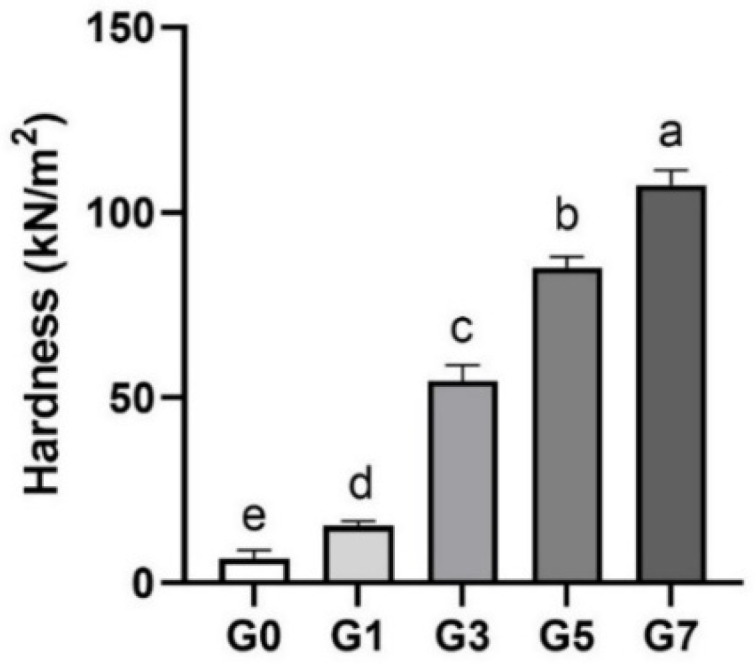
The hardness of abalone 3D print ink according to gelatin concentration (0–7%). Error bars represent the standard deviation, and different letters on the top of the column indicated a significant difference (*p* < 0.05).

**Figure 2 foods-11-03262-f002:**
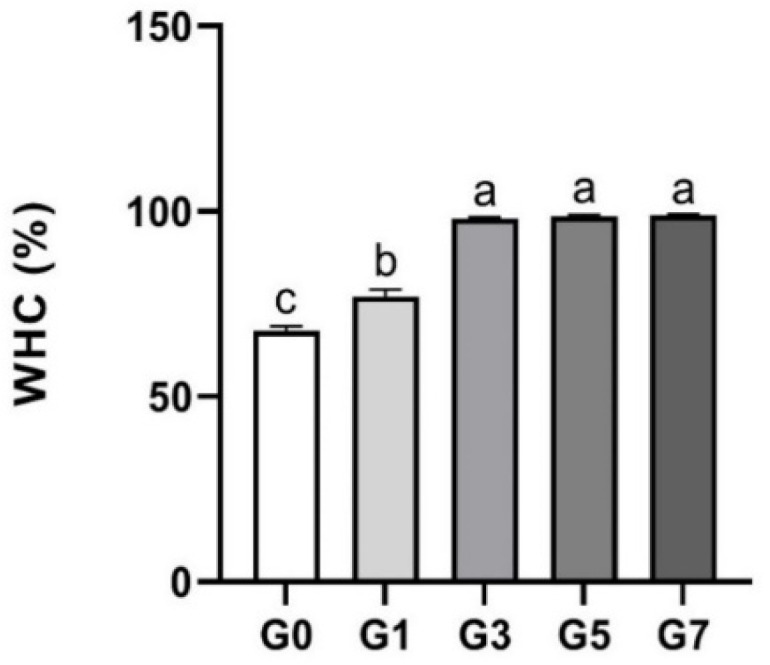
Water holding capacity of abalone 3D print ink at various concentrations of gelatin (0–7%). Error bars represent the standard deviation, and different letters on the top of the column indicated a significant difference (*p* < 0.05).

**Figure 3 foods-11-03262-f003:**
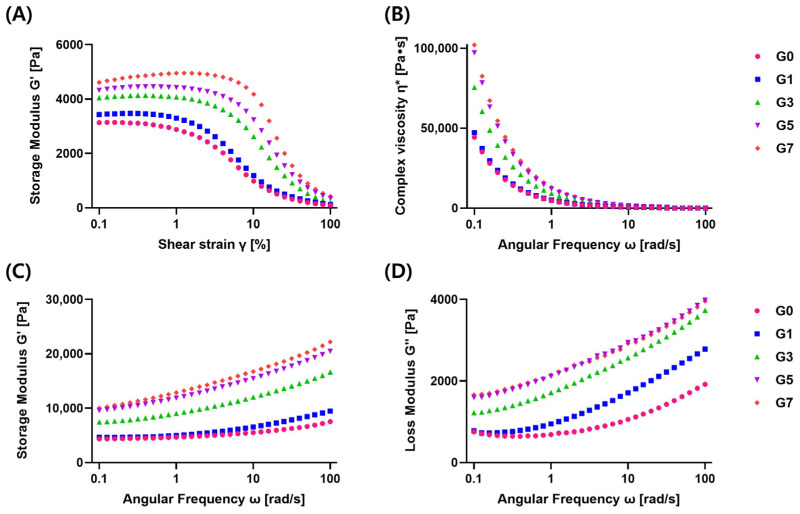
Linear viscoelastic region (LVE) during an amplitude sweeps test (**A**). Complex viscosity (η*), storage modulus (G′), and loss modulus (G″) during an angular frequency sweep test (**B**, **C**, and **D**, respectively) for abalone 3D print ink containing different concentrations of gelatin (0–7%).

**Figure 4 foods-11-03262-f004:**
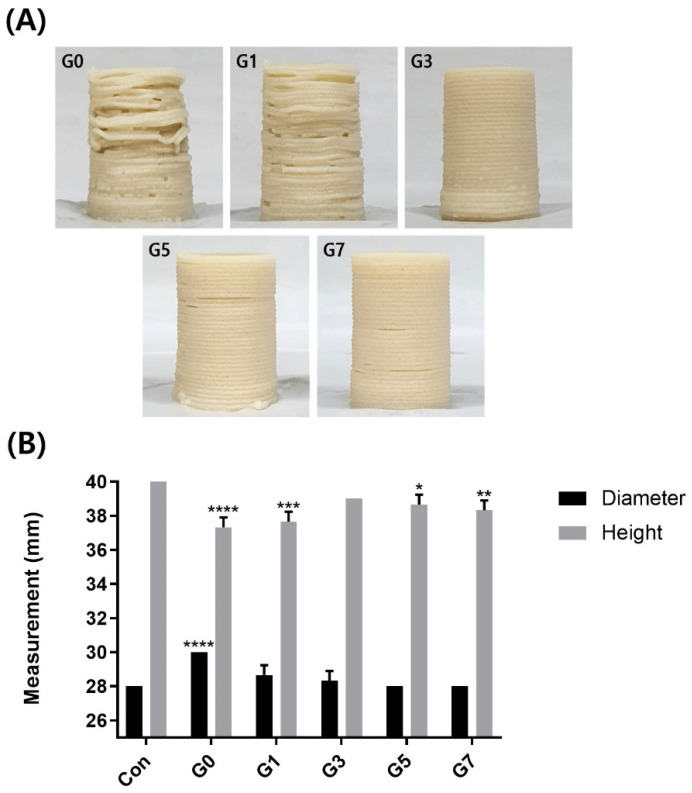
Cylindrical printing tests of Abalone 3D print ink with different concentrations of gelatin (G0, 0%; G1, 1%; G3, 3%; G5, 5%; G7, 7%) (**A**). Measurement of the diameter and height of cylindrical printing (**B**). Con indicated the original 3D model value (diameter of 28 mm, height of 40 mm). Error bars represent standard. *^, ^**^, ^***^, ^**** *p* < 0.05 vs. diameter and height of Con by two-way ANOVA.

**Figure 5 foods-11-03262-f005:**
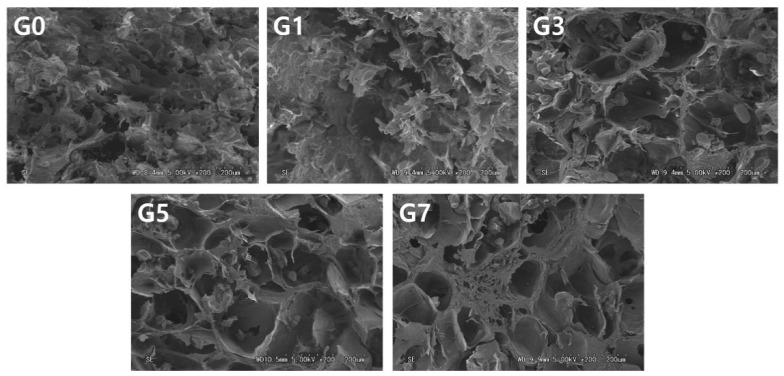
Scanning electron microscope images of abalone 3D print ink according to gelatin concentration (G0, 0%; G1, 1%; G3, 3%; G5, 5%; G7, 7%).

**Table 1 foods-11-03262-t001:** Nutritional compositions (g) per 100 g of the abalone 3D print inks (APIs).

Sample	Gelatin	Abalone	Isolated Soy Protein	Polydextrose	Vitamin C	Gellan Gum	Water
G0	0	10	4.5	2.5	0.01	1	82
G1	1						81
G3	3						79
G5	5						77
G7	7						75

## Data Availability

Data is contained within the article.

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
