# Peer review of "Development of an Abalone 3D Food Printing Ink for the Personalized Senior-Friendly Foods"

_foods, 2022, doi:10.3390/foods11203262_

Round 1

Reviewer 1 Report

The article entitled “Development of an abalone 3D food printing ink for the personalized senior-friendly foods” has been carefully reviewed. I have following comments on the article.

1.       Some people might be allergic to soya protein, suggest the author add some points and discuss this issue in the article.

2.       Dietary fiber is also very important for elder people, how the author can overcome this issue in 3D printing foods?

3.       Table 1. The author did not optimize the conditions (like abalone, Isolated soy protein, etc.) to develop 3D print inks. justify

4.       Did the author try to develop other shapes using 3D printing? Justify

5.       According to Fig.4 A, G7 look like the best-printed food, Why the author did not increase the gelatin content and go beyond 10 g to check up to which level gelatin can be added?  

6.       Author should prepare a graphical abstract of the article for better understanding. 

Author Response

The authors appreciate the reviewer’s kindly helpful comments that improve the quality of the manuscript. Revision has been extensively made throughout the whole manuscript to improve clarity following the reviewer’s comments. Also, the manuscript has been proofread by a professional English edition service. The changes are shown in red in the revised manuscript. The following responses have been prepared to address all of the reviewer’s comments in a point-by-point fashion

Reviewer 2 Report

L’article: « Développement d’une encre d’impression alimentaire 3D ormeau pour les aliments personnalisés adaptés aux personnes âgées » est clair et bien structuré. L’objectif est bien énoncé. Cependant, il n’y a pas de discussion globale sur les différents résultats obtenus. Le « niveau standard d’aliments gériatriques » est mentionné plusieurs fois, veuillez vous y référer.

L4 : Il manque une virgule après « Won-Kyo Jung ».

L62 : Les normes européennes considèrent la gélatine comme un ingrédient et non comme un additif. Si c’est également le cas en Corée, veuillez le préciser pour éviter toute confusion.

L63-65 : La phrase des lignes 63 à 65 me semble inutile. Il conviendrait de le supprimer ou de le développer davantage.

L68-72 : Des méthodes alternatives à l’utilisation d’additifs peuvent être envisagées pour limiter les formulations. (par exemple : Portanguen, S., Tournayre, P., Gibert, P., Leonardi, S., Astruc, T., & Mirade, P. S. (2022). Développement d’une imprimante 3D pour la fabrication de gels protéiques alimentaires fonctionnels. Aliments, 458).

L74 : S’agit-il de la référence 15 ou 15bis ? A vérifier selon le problème de pagination de L327.

L73-80 : L’ormeau est parfois rare dans certains pays. Veuillez préciser s’il s’agit d’une ressource abondante en Corée issue de l’agriculture. Donc, si ce travail peut être transféré à un niveau industriel de manière viable.

L81-86 : Veuillez ne pas écrire ce paragraphe au passé.

L92 : Veuillez indiquer le degré de floraison de la gélatine ainsi que son type (A ou B) et son origine (mammifère ou non, origine anatomique). Chaque gélatine a ses propres spécificités.

L94-102 : L’article 15 met en évidence certaines pertes nutritionnelles pendant la cuisson. Est-ce un critère que vous avez pris en considération? Comment les gammes de cuisson ont-elles été définies ?

Tableau 1 : Veuillez indiquer la teneur en eau des produits. L’eau représente une partie importante de ces aliments.

L117 : Veuillez indiquer comment la surface de l’échantillon a été détectée.

L119 : Il manque un point à la fin du paragraphe

L122 : Pouvez-vous expliquer pourquoi l’échantillon est si petit (500 mg) ? Est-il représentatif?

L139-150: La gélatine est normalement solide à température ambiante, veuillez indiquer ses caractéristiques à la section 2.3 comme mentionné ci-dessus. Tout contrôle de la température pendant la phase d’impression doit être indiqué. Veuillez également indiquer le débit.

L145 : Le taux de remplissage indiqué est de 0 %. Veuillez indiquer la distance entre les supports verticaux internes et la façon dont ils se croisent. Une vue transversale (une image ou une modélisation numérique serait appréciée).

L155: Veuillez vérifier le grossissement indiqué.

L162 : Il est difficile de relier les mesures de dureté à une réalité physiologique. Pourquoi ne pas utiliser le test TPA pour évaluer la mastication de l’aliment?

L168 : Comme indiqué dans la demande de la ligne 62, veuillez nous indiquer quelles normes sont touchées.

L177 : La comparaison avec le caoutchouc est mal choisie. Ces 2 matériaux sont très différents. Veuillez vous en tenir à des critères biologiques objectifs, en particulier en ce qui concerne le degré de floraison et la longueur des chaînes protéiques qui en dépendent.

L177-179 : Je ne comprends pas cette phrase, veuillez la reformuler. Encore une fois, la force d’un gel de gélatine devrait être liée à son degré de floraison, pas seulement à sa concentration.

L199: Le WHC n’a pas été réduit pour la condition G0 puisque c’est votre condition de contrôle? Dans l’affirmative, veuillez reformuler la phrase L198-200.

L206-208 : Veuillez indiquer à la section 2.6 si les mesures des propriétés viscoélastiques ont eu lieu avant ou après l’impression.

Fig. 3: Whenever possible, the curves for G' and G'' should be on the same graph.

L238-253: Were mass measurements made to verify that the extrusion rate was consistent between the different formulations?

L269: Is the term "cellular" the right one? To be reconsidered.

L270-271 : Vous parlez d’une augmentation de l’épaisseur de la paroi due à la gélatine. Ensuite, vous parlez d’amidon, je ne vois pas la cohérence.

L307 : Référence incorrecte/incomplète.

L325 : Problème de pagination.

L328 : Problème de pagination.

L334 : Référence incomplète.

Author Response

(The authors gave the same response as above.)
